# Alteration of Excitation/Inhibition Imbalance in the Hippocampus and Amygdala of Drug-Resistant Epilepsy Patients Treated with Acute Vagus Nerve Stimulation

**DOI:** 10.3390/brainsci13070976

**Published:** 2023-06-21

**Authors:** Qian Yi Ooi, Xiaoya Qin, Yuan Yuan, Xiaobin Zhang, Yi Yao, Hongwei Hao, Luming Li

**Affiliations:** 1National Engineering Research Center of Neuromodulation, School of Aerospace Engineering, Tsinghua University, Beijing 100084, China; 2Precision Medicine and Healthcare Research Center, Tsinghua-Berkeley Shenzhen Institute, Tsinghua University, Shenzhen 518071, China; 3Department of Functional Neurosurgery, Xiamen Humanity Hospital, Fujian Medical University, Fuzhou 350005, China; 4Surgery Division, Epilepsy Center, Shenzhen Children’s Hospital, Shenzhen 518038, China; 5IDG/McGovern Institute for Brain Research, Tsinghua University, Beijing 100084, China

**Keywords:** excitation/inhibition imbalance, vagus nerve stimulation, drug-resistant epilepsy, stereo-electroencephalography

## Abstract

An imbalance between excitation (E) and inhibition (I) in the brain has been identified as a key pathophysiology of epilepsy over the years. The hippocampus and amygdala in the limbic system play a crucial role in the initiation and conduction of epileptic seizures and are often referred to as the transfer station and amplifier of seizure activities. Existing animal and imaging studies reveal that the hippocampus and amygdala, which are significant parts of the vagal afferent network, can be modulated in order to generate an antiepileptic effect. Using stereo-electroencephalography (SEEG) data, we examined the E/I imbalance in the hippocampus and amygdala of ten drug-resistant epilepsy children treated with acute vagus nerve stimulation (VNS) by estimating the 1/f power slope of hippocampal and amygdala signals in the range of 1–80 Hz. While the change in the 1/f power slope from VNS-BASE varied between different stimulation amplitudes and brain regions, it was more prominent in the hippocampal region. In the hippocampal region, we found a flatter 1/f power slope during VNS-ON in patients with good responsiveness to VNS under the optimal stimulation amplitude, indicating that the E/I imbalance in the region was improved. There was no obvious change in 1/f power slope for VNS poor responders. For VNS non-responders, the 1/f power slope slightly increased when the stimulation was applied. Overall, this study implies that the regulation of E/I imbalance in the epileptic brain, especially in the hippocampal region, may be an acute intracranial effect of VNS.

## 1. Introduction

Neurodevelopmental encephalopathy is a complex condition that involves a disruption in the normal development and functioning of the brain. While an imbalance between excitatory and inhibitory (E/I) neuronal circuits is an essential aspect of this condition, it can have profound effects on brain structure and function, leading to cognitive, behavioral, and neurological impairments [1,2,3,4]. E/I imbalances are mainly due to defects in gamma-aminobutyric acid (GABA)-mediated activities and hyperexcitability caused by an increased glutamatergic signaling and function [5]. In a healthy condition, E/I in the brain is modulated to promote information flow and communication between remote functional regions [6]. Variations in the E/I ratio can result in a variety of neurological and mental diseases, including epilepsy [7,8], schizophrenia [9], and autism [10,11,12]. Although seizure mechanisms are complicated and their precise underpinnings are mostly unknown, it is hypothesized that seizures are primarily produced by abnormal interactions between excitatory and inhibitory cells, synchronization, and an abrupt aberrant firing of neurons [8,13,14,15,16,17,18]. While *Mecp2* protein is related to the regulation of glutamatergic synapses and the functioning of the GABAergic circuitry, animal studies found an altered E/I balance in the CA3 hippocampal pyramidal neurons and layer IV neurons of the barrel cortex in *Mecp2* knockout mice, leading to seizures [19,20,21]. Given that a disruption in the E/I balance contributes to epileptic convulsions, it is crucial to examine the anti-epileptic mechanisms of treatment administered to epilepsy patients. Specifically, it is intriguing to determine if the therapeutic options offered to epilepsy patients can accomplish anti-epileptic effects by correcting the E/I imbalance.

Vagus nerve stimulation (VNS) is a peripheral nerve stimulation technique developed by Dr. Jake Zabara to lessen or eliminate canine epileptic episodes [22]. The U.S. Food and Drug Administration (FDA) approved VNS for the treatment of drug-resistant epilepsy (DRE) in 1997, but its potential for further advancement is somewhat limited by its uncertain anti-epileptic mechanism, questionable curative effect, and substantial individual variability. Imaging and electrophysiological studies demonstrated that VNS may exert an anti-epileptic effect by altering the epilepsy patient’s cerebral blood flow, desynchronizing the brain network, and reducing the functional connectivity, electroencephalography (EEG) power, and interictal epileptiform discharges (IEDs) in certain brain regions, particularly the hippocampus and amygdala [23,24,25,26,27,28,29,30,31,32,33,34,35,36,37]. There is also mounting evidence that VNS exerts an anti-epileptic effect through modifying neuronal excitability in the hippocampus and amygdala. For instance, animal investigations revealed a decrease in the field excitatory postsynaptic potential (fEPSP) slope and hippocampus EEG power during VNS [36,37], which is consistent with the earlier observation that vagal afferent stimulation suppresses cortical EEG power in cats [38]. VNS has been demonstrated to have a substantial effect on the neurophysiology of the hippocampus, suggesting a decreased excitability of hippocampal neurons despite an enhanced synaptic transmission efficiency. Additionally, VNS altered neuronal activity in the amygdala and hippocampus via modulating a subset of the postsynaptic density (PSD) proteome [39]. It is worthwhile to investigate how the implantation of electrodes into the brain can facilitate the research of electrophysiological effects on the hippocampus and amygdala of epilepsy patients.

The stereo-electroencephalography (SEEG) technique created in the 1950s by French surgeons Bancaud and Talairach is a well-established, safe, and minimally invasive neurosurgical method for investigating cortical and subcortical locations in epileptic patients [40,41,42,43,44]. In European, Canadian, American, and Chinese epilepsy surgery, intracranial EEG is commonly used to identify the seizure onset zone (SOZ) and localize seizures [45,46,47]. It is considered as the gold standard for data quality as the source of brain activity can be detected precisely in the brain [48]. Using SEEG, hippocampal and amygdala signals with an extremely high signal-to-noise ratio can be obtained for investigation.

In the present work, the electrophysiological alteration in the hippocampi and amygdalae was studied by analyzing the SEEG data of ten DRE children applied with acute VNS of different stimulation parameters. We expected that activating the vagus nerve of epileptic patients would alleviate E/I imbalance in the brain.

## 2. Materials and Methods

### 2.1. Subjects

This study included ten DRE patients who underwent implantation of a vagus nerve stimulator at Xiamen Humanity Hospital and Shenzhen Children’s Hospital. There were 9 males and 1 female, with an age range between 2 and 10 years old. There were two good responders (≥50% reduction in seizure frequency), three poor responders (<50% reduction in seizure frequency), and five non-responders (no change in seizure frequency). All patients underwent complete pre-surgical evaluations, including SEEG at the hospital. Patients’ demographic, clinical, radiographic, and electrophysiological information were collected, as shown in (Appendix A). Patients were diagnosed as having DRE according to the practical clinical definition of the International Anti-Epilepsy Alliance (ILAE) [49,50], which is the persistence of seizures at final follow-up despite the use of at least two appropriate anti-seizure medications (ASMs). The study protocol was approved by the ethics committees of Xiamen Humanity Hospital (code: AF-SOP-029-01.0) and Shenzhen Children’s Hospital (code: 202009302). Informed consent was obtained from the children or their guardians for the purpose of this study.

### 2.2. Vagus Nerve Stimulation

The VNS device was implanted by neurosurgeons according to standard procedures [51]. The devices of patients were turned off at least 2 weeks prior to the SEEG electrodes implantation surgery. After confirming that the patients had no complications such as edema formation and intracerebral hemorrhage on the 2nd or 3rd day after implantation of SEEG depth electrodes, the acute stimulation was applied during wakefulness. ASMs were not consumed by patients during this study. The electrical stimulation was commenced from 0.8 mA to 2.2 mA with a signal frequency of 30 Hz, a pulse width of 500 μs, and a duty cycle of 22.5 % (14 s ON/1.1 min OFF; 2 s “soft-reboot” was included before and after VNS-ON). For every stimulation parameter, VNS was turned on for 12 min. A total of 12 to 15 min of washout period was applied before turning on VNS of the following stimulation parameter, as shown in Figure 1. The optimal stimulation amplitude for all patients was obtained according to the patients’ or guardians’ feedback after treatment or the last amplitude used before this study if their optimal amplitude was uncertain: 1.8 mA (Patients 1, 3, 4, 5, 6, 7, and 8), 2.2 mA (Patient 2), 1.3 mA (Patient 9), and 1.5 mA (Patient 10).

### 2.3. SEEG Electrode Implantation and Data Acquisition

As part of the epileptogenic zone localization and workup for DRE of various etiologies, patients underwent robot-guided SEEG. On the day of operation, patients were transported to the operating room under general anesthesia. Attached to the head frame was the Robotic Stereotactic Assistance (ROSA) platform’s base, and the robotic arm was utilized for semiautomatic laser-based facial recognition to register the patients to the image. After sterile preparation, draping, and a surgical pause, the process commenced with the robotic arm following each predetermined trajectory. The 2D and 3D views of planned SEEG electrode implantation were as shown in Figure 1. While inserting the SEEG electrodes, the loss of cerebrospinal fluid was avoided and the electrodes were ensured to be undamaged. As deformation of skull may affect the accuracy of electrode implantation, intra-operative management was performed very cautiously in patients, especially anesthesia procedure and cerebrospinal fluid temperature monitoring. SEEG electrodes were carefully inserted into the brain tissue and the intra-operative complications were rare. After obtaining adequate data for VNS application, the electrodes were removed in the operating room under anesthesia.

During data analysis, the EEGLAB toolbox [52] in MATLAB R2021b (MathWorks Inc., Natick, MA, USA) was used for filtering and segmentation. Bipolar montage was utilized to limit volume conduction effects and bias caused by common references [53]. The raw data from the left and right hippocampi and amygdalae were selected and band-pass filtered between 1 Hz and 125 Hz.

### 2.4. 1/f Power Slope Estimation

In neural data, aperiodic activity follows a 1/f distribution, with power dropping exponentially with increasing frequency. This component can be described by a $1/f^{x}$ function, where the *x* parameter, also known as the aperiodic exponent, depicts the pattern of aperiodic power across frequencies and is similar to the negative slope of the power spectrum when measured in log–log space [54]. In addition, the measurement of the scaling behavior of the power spectral density (PSD) has been proposed as a potential method for studying diseased disorders [55]. The power-law exponent (slope) of the power spectrum (1/f) can be used to estimate synaptic E/I ratios [56]. As an indicator of E/I imbalance, the antagonism of information flow between SOZ and non-SOZ was proposed [47]. In this study, as cumulative effects may have led to bias in the results, we eliminated this to the greatest extent possible by selecting the first five stimuli of every parameter for analysis. Using Welch’s method (2 s windows, 50% overlap) in MATLAB R2021b (MathWorks Inc., Natick, MA, USA), the PSD of the five VNS-BASE (18 s) and VNS-ON (18 s) SEEG signals were determined. The computed PSD for each epoch was then averaged over all epochs. The 1/f slopes from 1 to 80 Hz were then calculated using the FOOOF package (https://github.com/fooof-tools/fooof, accessed on 15 November 2022) [57]. In essence, the 1/f power slope was fitted as a function across the given spectrum range, and each oscillatory peak was uniquely modeled with a Gaussian function. The algorithm’s settings were as follows: peak width limits = [1,6], max n peaks = 10, min peak height = 0.5, peak threshold = 2, and aperiodic mode = ‘fixed’. In order to approximate the 1/f power slope with time, SEEG data for averaged VNS-BASE and VNS-ON were first segmented into 2 s intervals, and then power spectrum density was estimated for each segment using Welch’s method (1 s windows with 50% overlap). The parameters set were: peak width limits = [2,6], max n peaks = 10, min peak height = 0.5, peak threshold = 2, and aperiodic mode = ‘fixed’.

### 2.5. Statistical Analysis

We used paired *t*-test to compare the mean value of each patient’s 1/f power slope during VNS-BASE and VNS-ON. A significance level of 0.05 was set. Statistical analysis and graphing were performed using GraphPad Prism 8.0 (GraphPad Software, San Diego, CA, USA). Information of specific graphs are explained in the figure legends. Values are given as mean ± SEM.

## 3. Results

### 3.1. Effect of Different Stimulation Amplitudes on the Regulation of E/I Imbalance

The E/I ratio was shown to be inferable from the 1/f power slope on the basis of computer modeling, with a larger slope (exponent) corresponding to a stronger E/I imbalance. We looked at how different VNS settings affected the E/I imbalance in the hippocampus and amygdala. Figure 2 depicts change in the 1/f power slope in the left and right hippocampi and amygdalae for all patients.

For Patient 1 with a good responsiveness to VNS, the 1/f power slope decreased when different output currents of stimulation were used, and the decrease in the right hippocampus and left amygdala was the most when optimal stimulation was used. The 1/f power slope change in the left hippocampus and amygdala of Patient 2 did not show a similar pattern. Under stimulation with optimal amplitude, the 1/f power slope decreased in the left hippocampus but increased in the left amygdala. It was odd when Patient 5 was given 1.3 mA of VNS as the 1/f power slopes in the left and right hippocampi and right amygdala rose, whereas, when 22.5 mA was used, the decrease in the 1/f power slope was the greatest. In Patient 6, variable VNS output currents increased the 1/f power slope of the right hippocampus. Stimulation with 1.8 mA, which was the optimal stimulation amplitude for this patient, increased the 1/f power slopes in both left and right hippocampi. For Patient 10, the change in 1/f power slopes in the right hippocampus and left and right amygdalae from baseline was minimal.

### 3.2. Dynamic Change in E/I from VNS-BASE to VNS-ON

From Figure 2, we learned that the percentage change in the 1/f power slope from baseline in both left and right hippocampi was more remarkable, and we further studied how VNS affected the E/I imbalance acutely with time. The 1/f power slopes of patients under optimal stimulation amplitudes were estimated starting from 18 s before stimulation was applied to the end of stimulation. It can be noticed that every patient had different 1/f power slopes in different regions during baseline. For Patients 1 and 2, who are good VNS responders, when VNS was turned on, the 1/f power slope reduced with time (Figure 3 and Figure 4). The decrease was significant in the right hippocampus of Patient 1 (*p* = 0.004). Although there was no prominent change in the 1/f power slope with time in poor responders, it can be seen that the 1/f power slope was dynamically stable before and during VNS. For non-responders (Patients 6 and 10), there was a slight increase in the 1/f power slope during stimulation.

## 4. Discussion

While increasing the stimulation intensity of VNS can recruit more vagal nerve fibers and potentially improve its efficacy [58], the optimal output current for VNS therapy varies among patients. Previous studies have found that many patients respond to low levels of current (<1 mA) [59], while another study suggested that 1.61 mA is the target population-level output current for VNS therapy for epilepsy [60]. However, the precise relationship between VNS intensity and its impact on neurotransmitter systems in the brain remains poorly understood. This study aimed to investigate the effects of VNS intensity on E/I imbalance in the brain to improve treatment outcomes. Our findings suggest that different stimulation parameters had varying effects on E/I balance, and that even modest VNS (0.8 mA) impacted E/I balance. This result is consistent with an animal study that suggested that low-output current VNS increased the release of GABA in the hippocampus [61]. In addition, the subjective patient impression of VNS is highly dependent on stimulation parameters, notably stimulation intensity. Our results show that the stimulation amplitude that had the greatest effect on the E/I balance of some patients was not the optimal stimulation amplitude reported as optimal. While different output currents can influence the brain differently, selecting appropriate stimulation intensities based on patient tolerance and the acute effect of VNS may improve the treatment outcome.

VNS is believed to exert its anticonvulsant actions through the vagus afferent network [62], which includes the hippocampi and amygdalae as essential components. The relationship between epilepsy, hippocampi, and amygdalae is significant, and studies have shown that both regions are susceptible to producing seizure activity [63,64,65]. The present study investigated how VNS affects hippocampus and amygdala regions and found that the impact on the hippocampus was more pronounced. This may be due to the high density of α7 neuronal nicotinic acetylcholine receptors (nAChRs) in the hippocampus, which are involved in the regulation of N-methyl-D-aspartate (NMDA) and GABA receptor activation [66]. VNS may improve the E/I imbalance in the hippocampus by activating vagal nerve fibers and stimulating cholinergic projections, leading to an enhanced synaptic plasticity, the regulation of E/I imbalance, and a reduction in seizure activity. In addition, there were significant changes in glia–neuron interactions and a substantial rise in extracellular glutamate levels during the transition from a preictal state to a generalized seizure [67]. The impairment of or reduction in the function of glutamate transporters within astrocytes in the hippocampus, along with the disruption of intercellular communication among astrocytes via gap junctions, is a critical event in the development of epilepsy [68,69]. It is probable that the reduction in E/I imbalance achieved by VNS may be linked to the amelioration of neurotransmission. On the other hand, animal studies have suggested that both hippocampal and amygdalar neuropathology are common in epilepsy patients. In addition to hippocampal damage, a considerable proportion of patients exhibit extensive amygdalar neuropathology [70,71,72]. As patients included in this study did not achieve seizure freedom, it is anticipated that adequate VNS-mediated anti-epileptic effects and treatment results could not be attained without modulation of the E/I imbalance in both hippocampus and amygdala regions.

Left–right asymmetries are commonly observed in the neural systems of bilaterians, likely evolving to optimize their usage [73]. Various imaging studies have shown that VNS had different effects on the left and right hippocampi and amygdalae. For example, short-term VNS has been found to decrease cerebral blood flow in the right hippocampus [74], decrease the blood-oxygen-level-dependent (BOLD) signal in the left hippocampus and left and right amygdalae [75,76], and deactivate the right hippocampus in single photon emission computed tomography (SPECT) activation studies [77]. Animal studies have also shown that acute VNS affected glucose metabolism in the left hippocampus [78]. In Patient 1, a significant improvement in E/I imbalance was observed in the right hippocampus when the optimal stimulation amplitude of VNS was applied, but not in the left, whereas, in Patient 6, the increase in E/I was greater in the left hippocampus compared to that in the right. These findings suggest that VNS may not uniformly affect left and right brain regions, and the effects of VNS may vary depending on the individual and stimulation parameters. More research is needed to fully understand how VNS affects the left and right brain differently.

There are several limitations in this study worth noting. Our sample size was small, to begin with. Not all patients had electrodes implanted in all left and right hippocampi and amygdalae. More patients with electrodes implanted in four of these regions may provide comparative value. In addition, this study lacked a control group. It would aid in studying the VNS effects on epilepsy patients if we were able to determine how the 1/f power slope changes with time in healthy subjects. As epilepsy is caused not only by E/I imbalance but also by an extreme form of synchronous brain activity, measuring only the 1/f power slope may be insufficient. Investigating the other factors while comparing or combining the results may aid in further studying the acute affects of VNS on epilepsy treatment.

## 5. Conclusions

Using SEEG data, which is a safe and reliable method of electrophysiological evaluation in children with epilepsy and can substantially facilitate the investigation of the VNS regulatory mechanism, we explored the disruption of the E/I balance in epileptic patients undergoing VNS. Inferred from the 1/f power slope, the E/I ratio acted differently to VNS of different protocols. Under an optimal stimulation intensity, we observed a reduced E/I imbalance during VNS-ON in patients with enhanced VNS responsiveness. The E/I imbalance in the brain of patients who do not respond well to VNS was not regulated by VNS. VNS may produce an anti-epileptic effect by regulating the E/I imbalance in both hippocampi and amygdalae.

## Figures and Tables

**Figure 1 brainsci-13-00976-f001:**
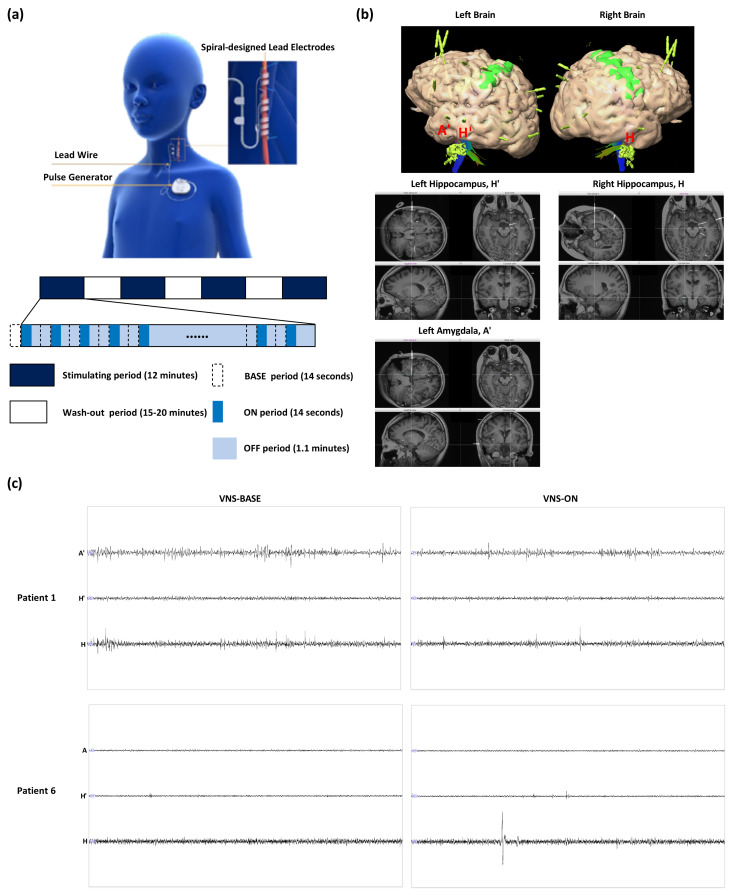
(**a**) VNS protocol, (**b**) 2D and 3D views of planned SEEG electrode implantation in Patient 1, and (**c**) SEEG signals of Patients 1 (left amygdala (A${}^{\prime }$), left hippocampus (H${}^{\prime }$) and right hippocampus (H)) and 5 (right amygdala (A), left hippocampus (H${}^{\prime }$) and right hippocampus (H)) during VNS-BASE and VNS-ON when 1.8 mA of stimulation amplitude was applied. VNS, vagus nerve stimulation. SEEG, stereo-electroencephalography.

**Figure 2 brainsci-13-00976-f002:**
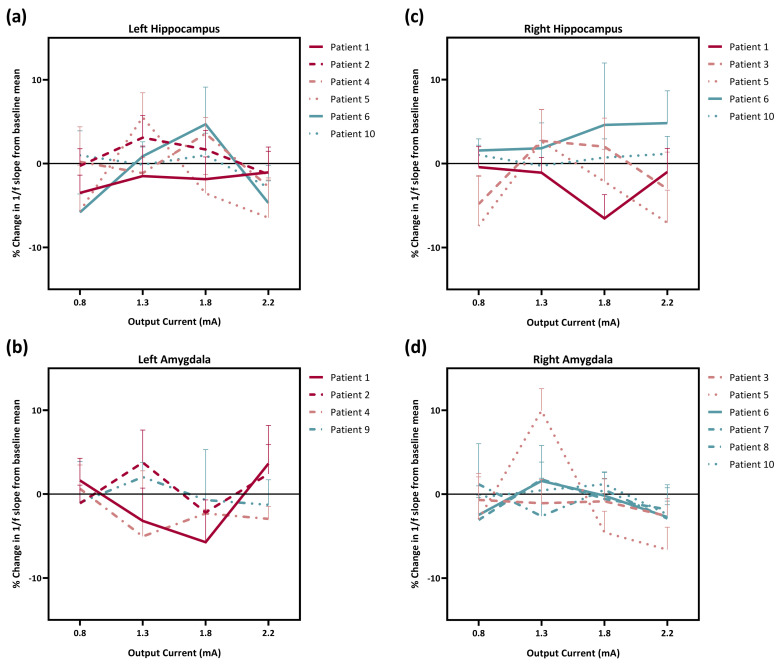
The percentage change $\left(\frac{Mean_{VNS-ON}-Mean_{VNS-BASE}}{Mean_{VNS-BASE}}\times 100\%\right)$ in 1/f power slope in the (**a**) left hippocampus, (**b**) left amygdala, (**c**) left hippocampus, and (**d**) right amygdala of VNS good responders (Patients 1 and 2), poor responders (Patients 3, 4, and 5), and non-responders (Patients 6, 7, 8, 9, and 10). VNS, vagus nerve stimulation.

**Figure 3 brainsci-13-00976-f003:**
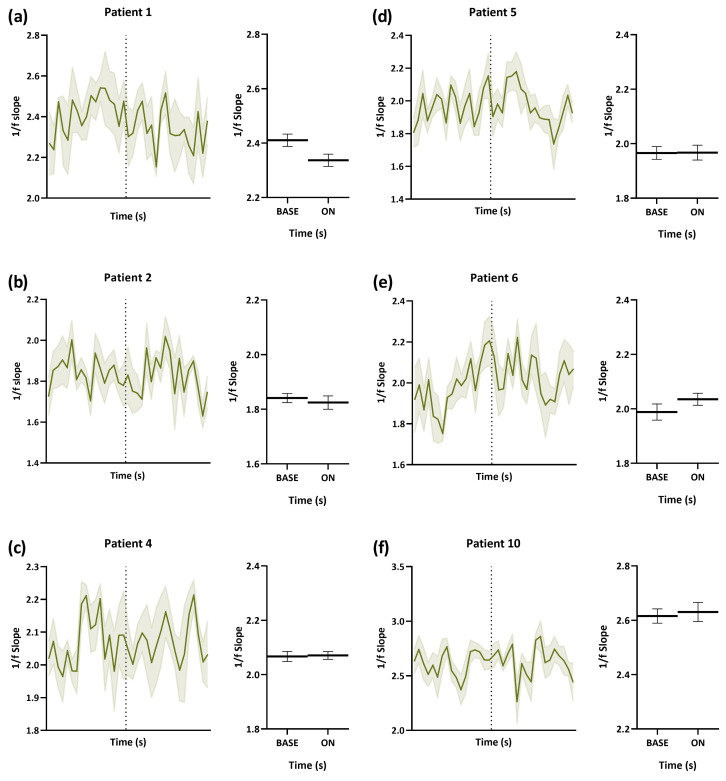
Time-varying E/I in the left hippocampus during VNS-BASE and ON. The 1/f power slopes of two good responders (Patients 1 and 2), two poor responders (Patients 4 and 5), and two non-responders (Patients 6 and 10) were estimated starting from 18 s before stimulation was applied to the end of stimulation. (**a**,**b**) The 1/f power slopes decreased when VNS was turned on. (**c**,**d**) The power slopes did not change. (**e**,**f**) There was an increase in the 1/f power slopes when VNS was turned on. E/I, excitation/inhibition. VNS, vagus nerve stimulation.

**Figure 4 brainsci-13-00976-f004:**
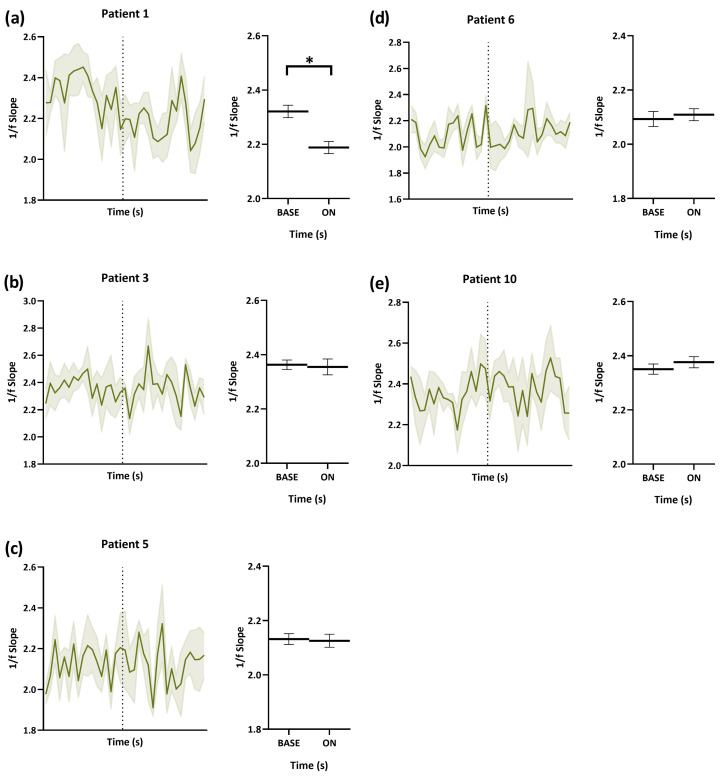
Time-varying E/I in the right hippocampus during VNS-BASE and ON. The 1/f power slopes of one good responder (Patient 1), two poor responders (Patients 3 and 5), and two non-responders (Patients 6 and 10) were estimated starting from 18 s before stimulation was applied to the end of stimulation. (**a**) There was significant decrease in the 1/f power slopes when VNS was turned on. (**b**,**c**) The power slopes did not change. (**d**,**e**) There was a slight increase in the 1/f power slopes when VNS was turned on. E/I, excitation/inhibition. VNS, vagus nerve stimulation. * *p* < 0.05.

## Data Availability

Anonymized data and documentation from this study can be made available to qualified investigators upon reasonable request. Such arrangements are subject to standard data sharing agreements.

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
