# Peer review of "Alteration of Excitation/Inhibition Imbalance in the Hippocampus and Amygdala of Drug-Resistant Epilepsy Patients Treated with Acute Vagus Nerve Stimulation"

_brainsci, 2023, doi:10.3390/brainsci13070976_

Round 1
Reviewer 1 Report
Comments and Suggestions for Authors
The authors present a interesting reseach evaluating the effect of VNS in the I/E imbalance of the brain. The research is sound and interesting and offers valuable data and information to epilepsy treatment researchers.
The manuscript is mature and I couln't find any modification to propose.
Author Response
Dear reviewer,
We appreciate you for your precious time in reviewing our paper. Thank you.
Sincerely,
Ooi Qian Yi
National Engineering Research Center of Neuromodulation,
School of Aerospace Engineering,
Tsinghua University,
Beijing, P. R. China
Reviewer 2 Report
Comments and Suggestions for Authors
Dear Authors,
The data You have collected are interesting and may shed some light on VNS mode of action.
Nevertheless, I have some questions and comments.
1. What was the clinical rationale of implementing SEEG in the patients?
2. What was the complication rate of SEEG implementation
3. Did the patient take any anti epileptic drugs during the study?
4. You have written that VNS device was switched off 2 weeks before SEEG implementation. How were patients protected from seizures during that period and how frequency of seizures changed in comparison with period when VNS was on?
5. The number of patients was rather small. Please, explain in more details generalizability of your conclusions?
6. Have you compared responders vs. non-responders? If so, what were the findings?
7. "Neurodevelopmental encephalopathy is primarily caused by an imbalance between 19
excitatory (E) and inhibitory (I) neuronal circuits in the brain" - that sentence is too much of simplification. Please, develop that.
Comments on the Quality of English LanguageENglish require some minor improvement with grammar, vocabulary and clearance of some sentences.
Author Response
Dear reviewer,
We appreciate you for your precious time in reviewing our paper and providing valuable comments. It was your valuable and insightful comments that led to possible improvements in the current version. The authors have carefully considered the comments and tried our best to address every one of them. We hope the manuscript after careful revisions meet your high standards. The authors welcome further constructive comments if any. Point-by-point responses are in the document attached.
Thank you.
Sincerely,
Ooi Qian Yi
National Engineering Research Center of Neuromodulation,
School of Aerospace Engineering,
Tsinghua University,
Beijing, P. R. China

Reviewer 3 Report
Comments and Suggestions for Authors
At the manuscript "Exploring the Acute Electrophysiological Effect of Vagus Nerve Stimulation on Patients with Drug-Resistant Pediatric Epilepsy Using Stereo-electroencephalography” by Drs. Qian Yi Ooi et al authors describing results of analysis of imbalance of excitation / inhibition (E/I) in the hippocampus and amygdala of drug-resistant epilepsy patients treated with acute vagus nerve stimulation (VNS) by estimating. Authors believe that this study implies that the regulation of E/I imbalance in the hippocampal region may be an acute intracranial effect of VNS.
The authors did a lot of research work and obtained interesting data. The technique and the experimental paradigm do not cause objections, as well as the substantiation of the conclusions, but I have some questions.
Describing the phenomena associated with Excitation/inhibition (E/I) balance, the authors presently rightly refer to mouse models, in particular to the MECP2 gene mutation that causes Rett syndrome in humans. Rett syndrome in humans is frequently associated with seizures, but as far as the hippocampal-cortical interaction is concerned, the situation is not so simple. It has been studied in models and has been described in recent papers. I would suggest using these:
Li-Jen Lee et al, Structural and functional differences in the barrel cortex of Mecp2 null mice ; J Comp Neurol; 2017; 15;525(18):3951-3961. doi: 10.1002/cne.24315.
and add a few words to the part of the manuscript related to Rett syndrome (Citation [10])
Although disruption of the E/I balance in epileptic patients undergoing VNS is beyond doubt, we would like to know the opinion of the authors on the mechanism of this phenomenon. Could the change in E/I balance be related to neuromodulation from the glial side? Indirect data on the participation of the gap junction in this process is:
Amiri Mahmood, Modified thalamocortical model: a step towards more understanding of the functional contribution of astrocytes to epilepsy ; J Comput Neurosci; 2012; 33(2):285-99. doi: 10.1007/s10827-012-0386-8.
Volnova et al, The Anti-Epileptic Effects of Carbenoxolone In Vitro and In Vivo; Int J Mol Sci; 2022; 8;23(2):663. doi: 10.3390/ijms23020663.
Carmen Diaz Verdugo et al; Glia-neuron interactions underlie state transitions to generalized seizures Nat Commun; 2019 Aug 23;10(1):3830. doi: 10.1038/s41467-019-11739-z.
I would suggest the author to pay attention to them and add something to the discussion.
Minor criticism:
1. I would add “excitation / inhibition balance” to Keywords
2. LINE 35: The U.S. Food and Drug Administration (FDA) approved VNS for the treatment of drug-resistant epilepsy (DRE) in 1997 and VNS in 2005…. - Probably a typo in this phrase - the meaning of the phrase is not clear
The presentation of a subject is systematic and comprehensive and analysis is proper.
Author Response

(The authors gave the same response as above.)

Round 2
Reviewer 2 Report
Comments and Suggestions for Authors
Dear Authors,
Thank You for your responses and clarifications. I am satisfied and I have no further comments.